# Vesicular Integral-Membrane Protein 36 Is Involved in the Selective Secretion of Fucosylated Proteins into Bile Duct-like Structures in HepG2 Cells

**DOI:** 10.3390/ijms24087037

**Published:** 2023-04-11

**Authors:** Mizuki Muranaka, Shinji Takamatsu, Tsunenori Ouchida, Yuri Kanazawa, Jumpei Kondo, Tsutomu Nakagawa, Yuriko Egashira, Koji Fukagawa, Jianguo Gu, Toru Okamoto, Yoshihiro Kamada, Eiji Miyoshi

**Affiliations:** 1Department of Molecular Biochemistry and Clinical Investigation, Osaka University Graduate School of Medicine, 1-7 Yamada-oka, Suita 565-0871, Osaka, Japan; 2Department of Pharmaceutics, School of Pharmaceutical Sciences, Health Sciences University of Hokkaido, 1757 Kanazawa, Toubetsu-cho, Ishikari-gun 061-0293, Hokkaido, Japan; 3Bio-Diagnostic Reagent Technology Center, Sysmex Corporation, 4-3-2 Takatsukadai, Nishi-ku, Kobe 651-2271, Hyogo, Japan; 4Division of Regulatory Glycobiology, Institute of Molecular Biomembrane and Glycobiology, Tohoku Medical and Pharmaceutical University, 4-4-1 Komatsushima, Aobaku, Sendai 981-8558, Miyagi, Japan; 5Institute for Advanced Co-Creation Studies, Research Institute for Microbial Diseases, Osaka University, 3-1 Yamada-oka, Suita 565-0871, Osaka, Japan; 6Department of Advanced Metabolic Hepatology, 1-7 Yamada-oka, Suita 565-0871, Osaka, Japan

**Keywords:** AFP, cargo protein, fucosylation, HepG2, lectin

## Abstract

Fucosylated proteins are widely used as biomarkers of cancer and inflammation. Fucosylated alpha-fetoprotein (AFP-L3) is a specific biomarker for hepatocellular carcinoma. We previously showed that increases in serum AFP-L3 levels depend on increased expression of fucosylation-regulatory genes and abnormal transport of fucosylated proteins in cancer cells. In normal hepatocytes, fucosylated proteins are selectively secreted in the bile duct but not blood. In cases of cancer cells without cellular polarity, this selective secretion system is destroyed. Here, we aimed to identify cargo proteins involved in the selective secretion of fucosylated proteins, such as AFP-L3, into bile duct-like structures in HepG2 hepatoma cells, which have cellular polarity like, in part, normal hepatocytes. α1-6 Fucosyltransferase (FUT8) is a key enzyme to synthesize core fucose and produce AFP-L3. Firstly, we knocked out the *FUT8* gene in HepG2 cells and investigated the effects on the secretion of AFP-L3. AFP-L3 accumulated in bile duct-like structures in HepG2 cells, and this phenomenon was diminished by *FUT8* knockout, suggesting that HepG2 cells have cargo proteins for AFP-L3. To identify cargo proteins involved in the secretion of fucosylated proteins in HepG2 cells, immunoprecipitation and the proteomic Strep-tag system experiments followed by mass spectrometry analyses were performed. As a result of proteomic analysis, seven kinds of lectin-like molecules were identified, and we selected vesicular integral membrane protein gene *VIP36* as a candidate of the cargo protein that interacts with the α1-6 fucosylation (core fucose) on *N*-glycan according to bibliographical consideration. Expectedly, the knockout of *the VIP36* gene in HepG2 cells suppressed the secretion of AFP-L3 and other fucosylated proteins, such as fucosylated alpha-1 antitrypsin, into bile duct-like structures. We propose that VIP36 could be a cargo protein involved in the apical secretion of fucosylated proteins in HepG2 cells.

## 1. Introduction

Glycosylation is a characteristic post-translational modification of proteins that occurs during birth, differentiation, carcinogenesis, and inflammation [1,2]. Fucosylation is one of the most important glycosylation processes in cancer and inflammation [3]. The levels of many fucosylated serum proteins are increased in patients with chronic liver diseases [4]. Alpha-fetoprotein (AFP) is a well-known biomarker for hepatocellular carcinoma, but a problem is that serum AFP levels are increased in patients with liver cirrhosis and chronic hepatitis [5]. In contrast, fucosylated AF, referred to as AFP-L3, is a specific biomarker for hepatocellular carcinoma (HCC) [6]. The greater than 90% specificity of AFP-L3 allows differential diagnosis of HCC from liver cirrhosis and chronic hepatitis [7]. In 2009, AFP-L3 was approved as a cancer biomarker by the FDA (Food and Drug Administration) in the USA [8] and is highly evaluated for HCC-specific biomarkers now [9,10,11,12,13]. To understand the molecular mechanisms underlying the increase in AFP-L3 levels in the sera of patients with HCC, we purified alpha-1-6-fucosyltransferase (FUT8) from the porcine brain and the conditioned medium of a human gastric cancer cell line [14,15]. Although FUT8 expression is increased in HCC and cirrhotic liver tissue, it is not expressed in normal human liver [16]. Therefore, we suggest that another mechanism that does not involve increases in FUT8 expression in the liver could explain increased AFP-L3 levels in the sera of patients with HCC. In contrast, core fucose synthesized by the reaction of FUT8 is very important for cell proliferation and hepatocarcinogenesis [3,4].

Fucosylation is regulated by a variety of fucosylation-regulatory genes, including fucosyltransferases [3]. Guanosine diphosphate (GDP)-fucose, a donor substrate of fucosyltransferases, and GDP-fucose transporter are required to produce fucosylated proteins [17]. We previously reported that the expression of GDP-fucose transporter and FX, a rate-limiting enzyme of GDP-fucose, was dramatically increased in human HCC tissue compared with surrounding cirrhotic liver tissue [18]. In contrast, secretion into the bile duct of fucosylated proteins produced by hepatocytes was suppressed in *Fut8*-deficient mice, which suggests that selective secretion of fucosylated proteins occurs in the normal liver [19]; this process might be deficient in HCC tissue. Selective increases in fucosylated proteins in serum and bile are also observed in animal models of hepatocarcinogenesis [20]. Reduction in the expression of fucosylation-regulatory genes decreases AFP-L3 levels in bile duct-like structures of HepG2 hepatoma cells, which have cellular polarity [21]. These data suggest the presence of cargo proteins that regulate the secretion of fucosylated proteins into the apical side of the bile duct in hepatocytes and HepG2 cells.

We have investigated such lectin-like cargo proteins for over 10 years using a variety of glycoproteomic approaches. There is plenty of evidence that shows apical secretion of certain types of fucosylated glycoproteins in polarized cells; however, no specific candidates have been reliably identified. Recently, we created a novel anti-glycan antibody, a fucosylated AFP-specific monoclonal antibody (FasMab), that directly recognizes AFP-L3 [22]. FasMab is a powerful tool to more accurately confirm the presence of AFP-L3. Recent advances in protein–protein interaction analysis, such as the Strep-tag/Strep-Tactin system, have opened up the possibility of identifying novel lectin-like cargo proteins. In this present study, we identified seven kinds of lectin-like molecules with the Strep-tag/Strep-Tactin system and selected the vesicular integral-membrane protein gene 36 (VIP36) as a cargo protein for core-fucosylated *N*-glycans according to a previous paper of a simulation study [23]. Then, we investigated the possibility that VIP36 regulates the selective secretion of fucosylated proteins into bile duct structures of HepG2 cells.

## 2. Results

### 2.1. Immunostaining of AFP-L3 Was Not Observed in FUT8-Knockout HepG2 Cells

Because HepG2 cells form a bile duct-like structure, the accumulation of AFP-L3 in this structure was examined by immunostaining using an antibody against MRP2, a marker of bile ducts. Co-staining of AFP and MRP2 was observed in WT HepG2 cells but not in *FUT8*-KO HepG2 cells, and cytosolic staining of AFP was widely observed in both cell types (Figure 1A). Similar staining patterns were not observed in the HCC cell line Huh7, which does not form bile duct-like structures. Positive staining of AFP-L3 with FasMab was detected in the bile duct-like structures of WT HepG2 cells and not in the bile duct-like structures of *FUT8*-KO HepG2 cells (Figure 1B).

### 2.2. Identification of Cargo Receptors for AFP-L3 by Immunoprecipitation

As mentioned earlier, HepG2 cells probably have a system that transports AFP-L3 to the bile duct-like structures. We hypothesized the existence of a lectin-like cargo protein receptor for AFP-L3. To identify such cargo receptors, immunoprecipitation with an anti-AFP antibody was performed on both a HepG2 cell-conditioned medium and cell lysates to investigate the involvement of a possible cargo receptor of core fucose.

As shown in Figure 2A, there was no difference in the silver-staining pattern between WT and *FUT8*-KO HepG2 cells in the cell culture supernatant. After immunoprecipitation of cell lysates with FasMab, silver staining increased specific low molecular weight bands in WT HepG2 cells and an approximately 32 kDa MW band in *FUT8*-KO HepG2 cells (Figure 2B). Unfortunately, these proteins could not be identified by mass spectrometry analysis. Although we have tried immunoprecipitation experiments with various conditions for many years, we could not find any candidates of cargo proteins for fucosylated AFP. Cargo proteins involved in membrane trafficking were expected to be abundant in the endosomal fraction.

Next, to obtain an endosomal fraction of HepG2 cells, a sucrose density gradient ultracentrifugation method was performed to know whether or not intracellular localization/sorting of glycoproteins are changed with/without FUT8. As shown in Figure 3, immunoblot analysis showed positive bands for LAMP-1 (representative endosome marker) and AFP in fractions 3 and 4 in WT HepG2 cells. In contrast, these positive bands were only observed in fraction 4 in *FUT8*-KO HepG2 cells. These data suggest that glycoprotein localization or the intracellular transport system is slightly different between WT and *FUT8*-KO HepG2 cells. If there is a cargo receptor for fucosylated proteins, it is possibly condensed in fraction 4. However, immunoprecipitation of fraction 4 proteins with anti-AFP antibody followed by silver staining did not show different bands between WT and *FUT8*-KO HepG2 cells.

### 2.3. Proteomics Analysis of AFP Using the Strep-Tag System

Although the binding of lectins to glycans is much weaker than that of antibodies to antigens, immunoprecipitation may not be able to isolate lectin-like cargo protein receptors that target the core fucose. Furthermore, contamination of immunoprecipitation samples by nonspecific binding products other than lectin and glycoprotein binding products is unavoidable. Therefore, we used the Strep-tag/Strep-Tactin system in this study to isolate proteins bound to AFP. Proteins bound to AFP in WT and *FUT8*-KO HepG2 cells were electrophoresed on a 10% SDS-polyacrylamide gel and then silver-stained (Appendix A). For unknown reasons, total protein levels were slightly higher in *FUT8*-KO cells than in WT HepG2 cells. Mass spectrometry analysis identified 1043 proteins. Of these, 55 proteins were identified in *FUT8*-KO HepG2 cells only, 91 proteins were identified in WT HepG2 cells only, and 897 proteins were identified in both WT and *FUT8*-KO HepG2 cells (Appendix A). All results of mass spectrometry analysis were shown as Supplemental MS data. We summarized the results in Figure 4C, which shows specific proteins that had a high number of hits or belong to the lectin family of proteins or the family of bile duct structural proteins. The number of mass spectrometry hits for AFP was remarkably high in both the WT and *FUT8*-KO HepG2 cells. Interestingly, we also identified the well-known glycan biomarkers alpha-1 antitrypsin and Mac-2 binding protein (M2BP). Interestingly, seven kinds of lectin-like molecules were identified with this proteomic analysis (Appendix A).

### 2.4. VIP36 Knockout Affects the Transport of AFP into the Bile Duct-like Structures of HepG2 Cells

Among the seven lectin-like molecules, we selected the *VIP36* gene, which has been reported as a cargo receptor-like protein for core fucose [23]. To verify VIP36 as cargo receptors for fucosylated proteins, we established VIP36 knockout HepG2 cells. As shown in Figure 4A, VIP36 protein expression was dramatically reduced by *VIP36* KO, even though the cells did not establish a stable cell line. The MRP2 staining pattern has very little change in *VIP36*-KO HepG2 cells compared with WT HepG2 cells (Figure 4B). As expected, little AFP staining of bile duct-like structures was observed in *VIP36*-KO HepG2 cells (Figure 4B). Thirty microscopic fields were randomly selected, and two people independently and blindly determined whether the staining was positive or negative for AFP at the bile duct-like structures (Figure 4C,D). Approximately 28% of bile duct-like structures were positive for AFP in WT HepG2 cells, and this was reduced by half in *VIP36*-KO HepG2 cells (Figure 4D).

### 2.5. Effect of VIP36 Knockout Affects the Alpha-1 Antitrypsin Localization in Bile Duct-like Structures of HepG2 Cells

To determine the relationship of fucosylated glycoproteins other than AFP and VIP36 on protein sorting, we evaluated the staining of alpha-1 antitrypsin in the bile duct-like structures of HepG2 cells. Selective secretion of fucosylated alpha-1 antitrypsin was reported in our previous mouse study [19]. Similar to the AFP results, the percentage of *VIP36*-KO HepG2 cells with anti-alpha-1 antitrypsin staining in bile duct-like structures was about half that of WT HepG2 cells (Figure 5A). Next, we compared the inhibitory effects of *VIP36*-KO and *FUT8*-KO on alpha-1 antitrypsin localization in bile duct-like structures of HepG2 cells. Alpha-1 antitrypsin sorting to bile duct-like structures was suppressed more by *VIP36* KO than by *FUT8* KO (Figure 5B,C), suggesting that VIP36 affects the secretion of other fucosylations except for core fucose.

## 3. Discussion

For over 15 years, we have searched for cargo receptors for fucosylated proteins in hepatocytes and the liver. However, we failed to identify such lectin-like molecules using glycan affinity chromatography. *Fut8*-deficient mice have low levels of alpha-1 acid glycoprotein and alpha-1 antitrypsin in bile, although these glycoproteins are abundant in the blood of *Fut8*-deficient mice [19]. Cell fractionation of WT and *FUT8*-KO HepG2 cells showed different patterns of glycoproteins within endosomes (Figure 3). This data strongly implies there is a selective secretion of fucosylated glycoproteins in the liver. The recently established Strep-tag/Strep-Tactin system enables the detection of intracellular expression of a tag-labeled target protein, and coprecipitating molecules can interact with the protein of interest using beads that strongly bind to the tag. This system reduces noise, as it is not affected by the nonspecific adsorption of antibodies. Using this method, we identified approximately 200 proteins anticipated to interact with AFP in WT and *FUT8*-KO HepG2 cells (Appendix A). Interestingly, we detected more glycoproteins in the bile duct-like structures in WT HepG2 cells than in *FUT8*-KO HepG2 cells, suggesting that the core fucose is involved in the selection of bile duct-related proteins to be secreted (Appendix A). In fact, the bile duct-like structures in *FUT8*-KO HepG2 cells changed in shape or size. A variety of lectin-like molecules were also identified by anti-AFP immunoprecipitation followed by mass spectrometry. Among the seven kinds of lectin-like molecules, VIP36 was reported as a cargo protein for the core fucose on *N*-glycans, using simulation analysis [23]. A previous study showed VIP36 mainly recognizes high-mannose glycans, which are insufficient for oligosaccharide synthesis [24]. However, lectin–glycan interaction is influenced by the structure of glycoproteins, and it is possible VIP36 can recognize two types of glycans. We found that *VIP36* KO inhibited the selective sorting of alpha-1 antitrypsin more than AFP, suggesting that VIP36 recognizes terminal fucosylation as well as core fucose. In contrast, *FUT8* KO completely inhibited the selective sorting of AFP more than alpha-1 antitrypsin. These differences may depend on the type of fucosylation linkage or the molecular structure of the glycoproteins involved.

We were unable to conclusively determine VIP36 as a cargo receptor for fucosylated proteins with conventional affinity chromatography and immunoprecipitation possibly due to the low affinity of VIP36 for fucosylated glycoproteins. Lectin–glycan interaction is fragile compared to protein–protein interaction, and lectin–glycan interaction is influenced by pH change, temperature, and salt concentration. The graphic abstract represents our hypothesis that VIP36 plays an important role in the intracellular transport of glycoproteins, which might be pH dependent. Each organelle has a different homeostatic pH, so the binding affinity of VIP36 for fucosylated proteins may be dependent on intracellular and intra-organelle pH levels. VIP36 dysfunction from genetic mutations might cause cholestatic disease of unknown etiology in humans.

In summary, VIP36 is a candidate protein that might be involved in the sorting of fucosylated proteins into the apical bile duct and might be also involved in the pathogenesis of certain liver diseases.

## 4. Materials and Methods

### 4.1. Cell Lines

The wild-type (WT) human hepatoma cell line HepG2 was obtained from American Type Culture Collection. *FUT8*-knockout (*FUT8*-KO) HepG2 cells were established previously [25]. These cells were cultured in Dulbecco’s Modified Eagle Medium (#08456-36, Nacalai Tesque, Kyoto, Japan) supplemented with 10% fetal bovine serum (#SH30910.02, HyClone, Logan, UT, USA) and 100 units/mL penicillin and 100 μg/mL streptomycin (#09367-34, Nacalai Tesque) at 37 °C under 5% CO_2_ in humidified air. The human HCC cell line Huh7 was purchased from RIKEN BioResource Research Center and cultured in RPMI 1640 (#30264-56, Nacalai Tesque) supplemented with 10% fetal bovine serum and 100 units/mL penicillin and 100 µg/mL streptomycin at 37 °C under 5% CO_2_ in humidified air.

### 4.2. Immunohistochemistry

WT HepG2, *FUT8*-KO HepG2, and Huh7 cells were inoculated into a 35 mm diameter glass-bottom dish (#3961-035, IWAKI, Iwaki, Japan) and cultured for 3 days until approximately 80% confluency was reached. After removing the culture medium, the cells were washed with 1 mL phosphate-buffered saline (PBS) without calcium and magnesium [PBS (−)], and the cells were fixed in 1 mL 4% paraformaldehyde (#26126-54, Nacalai Tesque) at room temperature for 10 min. Then, the paraformaldehyde was removed, and the cells were washed with 1 mL PBS (−) for 5 min. To permeate the cells, the PBS (−) was replaced with 0.1% Triton X-100 (#807426, MP Biomedicals, Seven Hills, Sydney) in PBS (−) at room temperature for 30 min. The cells were then washed 3 times with 1 mL PBS (−) and then treated with 150 mM dithiothreitol (#14112-94, Nacalai Tesque) in PBS (−) with 0.1% Tween 20 at room temperature for 30 min. After 3 washes with 1 mL PBS (−), a primary antibody diluted in PBS (−) with 0.1% Tween 20 supplemented with 10% normal donkey serum (#D9663, Sigma-Aldrich, St. Louis, MO, USA), 1% bovine serum albumin (#01863-35, Nacalai Tesque), and 0.3 M glycine (#077-00735, FUJIFILM Wako Pure Chemical Corporation, Richmond, VA, USA) was added, and the mixture was reacted overnight at 4 °C. The primary antibodies used were 8 μg/mL anti-AFP or 20 μg/mL anti-AFP-L3 antibodies (Sysmex Corporation [22], Hyogo, Japan) or anti-multidrug-resistance-associated protein 2 (MRP2) antibody (clone M_2_III-6, KAMIYA BIOMEDICAL, Tukwila, WA, USA) diluted 100-fold. After 3 washes with 1 mL PBS (−), the cells were incubated with 10 μg/mL Alexa Fluor 488-labeled donkey anti-mouse immunoglobulin G (#A21202, Invitrogen, Waltham, MA, USA) or Alexa Fluor 546-labeled donkey anti-rabbit immunoglobulin G (#A10040, Invitrogen) dissolved in 1% bovine serum albumin in PBS (−) and with 5 mg/mL 4′,6-diamidino-2-phenylindole (#D1306, Invitrogen) diluted 500-fold with PBS (−) containing 1% bovine serum albumin at room temperature for 60 min in the dark. Then, the cells were washed with PBS (−) three times, and fluorescent images were detected with a confocal microscope (FLUOVIEW FV10i, OLYMPUS, Tokyo, Japan). For nuclear staining of cells, 5 mg/mL 4′,6-diamidino-2-phenylindole diluted 400-fold in 1% bovine serum albumin and 0.3% Triton X-100 in PBS (−) was incubated with the cells at room temperature for 60 min in the dark. For anti-alpha-1 antitrypsin antibody staining, cells were not treated with dithiothreitol. Anti-alpha-1 antitrypsin rabbit polyclonal antibody (#16382-1-AP, Proteintech, Rosemont, IL, USA) was used at 5 μg/mL.

### 4.3. Immunoprecipitation

Recovery of cell culture supernatant: WT and *FUT8*-KO HepG2 cells were inoculated onto a 100 mm diameter dish and cultured to 70–80% confluency. Then, the medium was replaced with a serum-free medium, and after 2 days of culture, the supernatant was collected. The supernatant was centrifuged at 300× *g* for 10 min to remove cell debris and then filtered through a 0.45 μm filter. The filtered product was further concentrated with a 10 kDa Amicon Ultra-15 Centrifugal Filter (#UFC9010, Millipore, Burlington, MA, USA).

Recovery of whole-cell lysate: Cells were prepared in the same manner as for supernatant recovery. After removing the cell culture medium and washing 3 times with 1 mL PBS (−), 1 mL PBS (−) was added, and the cells were collected with a cell scraper. The sample was centrifuged at 200× *g* for 5 min at 4 °C, the supernatant was removed, 100 µL Golgi solution (10 mM 2-(N-morpholino)ethanesulfonic acid, 500 mM NaCl, 1% NP-40, 1 mM CaCl_2_, 0.1 mM MgCl_2_, and 0.1 mM MnCl_2_) containing a protease inhibitor cocktail (#25955-11, Nacalai Tesque) was used to resuspend the cell pellet, and a Bioruptor (Tosho Electric, Hakodate, Japan) was used to crush the cells ultrasonically for 5 min on ice with a 15 s on/off cycle. Then, the cell lysate was centrifuged at 20,000× *g* for 20 min at 4 °C, and the supernatant was collected as the whole-cell lysate.

### 4.4. Preparation of Endosome Fraction

The endosome fraction was prepared as follows, with reference to the method of de Araujo et al. [26]. In brief, WT and *FUT8*-KO HepG2 cells were seeded in 150 mm diameter dishes and cultured until 70–80% confluency was reached. The supernatant was removed, the cells were washed 3 times with 5 mL PBS (−), and then 5 mL PBS (−) was added, and the cells were collected with a cell scraper. The cells were centrifuged at 200× *g* for 5 min at 4 °C, the supernatant was removed, and the cells were resuspended by pipetting in HB solution (250 mM sucrose, pH 7.4, with protease inhibitor cocktail) at approximately 3 times the amount of the cell pellet. After centrifugation at 1300× *g* for 10 min at 4 °C, the supernatant was removed. Then, the cell pellet was again resuspended by pipetting in HB solution at approximately three times the amount of the cell pellet. The cell suspension was collected in a syringe equipped with a 22 G needle, and the cells were disrupted by syringing 30 times. An equal volume of HB solution was added, and the supernatant was collected by centrifugation at 2000× *g* for 10 min at 4 °C. A mixture of 21 mg WT or *FUT8*-KO HepG2 cell lysate with a 1.2-fold volume of 62% sucrose solution was placed in an ultracentrifuge tube (#344059, Beckman Coulter, Brea, CA, USA). A 1.5-fold volume of 35% sucrose solution was added, and then a 1-fold volume of 25% sucrose solution was layered above. Finally, the tube was filled to the top with HB solution. Fractionation was performed using an SW 41 Ti swinging-bucket rotor (Beckman Coulter) at 210,000× *g* for 90 min at 4 °C in an Optima XPN-90 ultracentrifuge (Beckman Coulter). The endosome fraction was then identified by immunoblotting with anti-lysosomal-associated membrane protein-1 (LAMP-1) antibody (#ab24170, Abcam, Boston, MA, USA) using 12 µL from each successive 1 mL aliquot collected from the upper end to the lower end of the ultracentrifuge tube. The membrane reacted with the anti-LAMP-1 antibody and was washed 3 times with TBST buffer and then reacted with a 5000-fold diluted HRP-conjugated anti-rabbit IgG antibody (#W4011, Promega, Alexandria, NSW, Australia). After washing three times with TBST buffer, band signals were detected using an ECL system (Chemi-Lumi One Super, #02230, Nacalai Tesque).

### 4.5. The Proteomic Strep-Tag System

The human AFP gene was obtained by polymerase chain reaction (PCR) cloning. Template cDNA was prepared from RNA extracted from HepG2 cells. PCR reactions were performed using the forward primer 5′-GCTTCCACCACTGCCAATAAC-3′ and the reverse primer 5′-CTCGTTTTGTCTTCTCTTCCCC-3′ under the following conditions: denaturation at 95 °C for 3 min; then 40 cycles of denaturation at 98 °C for 10 s, annealing at 56 °C for 15 s, and extension at 68 °C for 2 min; and a final extension at 68 °C for 7 min. The amplification product was electrophoresed and extracted from the agarose gel using the NucleoSpin Gel and PCR Clean-up mini-kit (#740609, MACHEREY-NAGEL, Düren, Germany). The purified product was treated with Taq DNA polymerase to add adenine to the C-terminal, and the product was subcloned into the pGEM-T Easy Vector System (#A1360, Promega) using TA cloning. The sequence of the cloned AFP gene was verified by DNA sequencing.

The pCAGGS-AFP-FOS vector was prepared as follows. The pCAGGS-FOS vector (kindly provided by Dr. Okamoto at Osaka University) was excised with EcoRI (#1040A, Takara Bio, Shiga, Japan) and purified. In order to integrate the AFP gene into the pCAGGS-FOS vector in the correct orientation, PCR was used to generate an amplification product with a linker containing the EcoRI sequence of the pCAGGS-FOS vector and vector sequences flanking the EcoRI sequence added to both ends of the AFP gene. PCR reactions were performed using the forward primer 5′-CTCATCATTTTGGCAAAGAATTCACCATGAAGTGGGTGGAATC-3′ and the reverse primer 5′-CCATGCATCGATGAGCTCGAATTCTAACTCCCAAAGCAGCACG-3′ under the following conditions: denaturation at 95 °C for 3 min; then 40 cycles of denaturation at 98 °C for 10 s, annealing at 56 °C for 15 s, and extension at 68 °C for 2 min 20 s; and a final extension at 68 °C for 7 min. The amplified product was electrophoresed and then extracted from the agarose gel using the NucleoSpin Gel and PCR Clean-up mini-kit. The purified product was incorporated into the pCAGGS-FOS vector cleaved by EcoRI by a recombination method using NEBuilder HiFi DNA Assembly Master Mix (#E2621, New England BioLabs, Ipswich, MA, USA).

### 4.6. Mass Spectrometry and Immunocytechemistry

Immunoprecipitated samples were separated by polyacrylamide gel electrophoresis and stained with a Silver Stain MS Kit (FUJIFILM Wako Pure Chemical Corporation) in accordance with the manufacturer’s instructions. Mass spectrometry analysis of silver-stained bands of interest that varied between WT and *FUT8*-KO HepG2 cells was performed by the Joint Research Center for Medical Research and Education at Osaka University, Osaka, Japan. To confirm our mass spectrometry data, WT and *VIP36*-KO HepG2 cells were treated with mouse anti-MRP2 and rabbit anti-AFP antibodies to analyze the localization of AFP in the bile duct-like structures by confocal microscopy (FLUOVIEW FV10i, OLYMPUS). Visualization with fluorescent labeling was performed using secondary antibodies as described in the immunohistochemistry section.

### 4.7. Production of VIP36 Knockout Vector 

The human *VIP36*-KO vector was prepared as follows. The purified products obtained from the BbsI (#R0539, New England Biolabs)-digested pX330-U6-Chimeric_BB-CBh-hSpCas9 vector (Plasmid #42230, Addgene, Watertown, MA, USA) that contains a puromycin-resistance gene were ligated with h*VIP36* sgRNA-forward 5′-caccgCAGGGGTCAGACGTACGTAC-3′ and h*VIP36* sgRNA-reverse 5′-aaacGTACGTACGTCTGACCCCTGc-3′.

### 4.8. Transfection of AFP Gene, VIP36 KO Vector and Immunoblotting of VIP36

WT HepG2 cells of 2.2 × 10^6^ were seeded onto collagen-coated 100 mm diameter dishes (#4020-010, IWAKI). The cell culture medium was changed 3 h before transfection. To form transfection complexes, 8 μg pCAGGS-AFP-FOS vector DNA and 24 μL GenJet In Vitro Transfection Reagent (#SL100489-HEPG2, SignaGen Laboratories, Frederick, MD, USA) were each diluted in 500 μL medium and then combined and allowed to stand for 20 min. The entire volume was added dropwise to a 100 mm diameter dish, and the medium was changed 18 h later. After another 24 h of culture, the cells were collected for immunoprecipitation. *VIP36*-KO HepG2 cells were similarly transfected, and 24 h after transfection, the complexes were replaced with medium containing 2 μg/mL puromycin (#anti-pr-1, InvivoGen, Toulouse, France) to establish a stably transfected cell line. KO of the *VIP36* gene was verified by immunoblotting with an anti-VIP36 antibody (#12455-MM02, Sino Biological, Beijing, China) using cell lysates extracted from the established stable cell line. 

### 4.9. Statistical Analysis

Statistical analyses were performed with an unpaired, 2-tailed Student *t*-test by using JMP Pro 16.0 software (SAS Institute Inc., Cary, NC, USA). *p* < 0.05 was considered to be statistically significant.

## Figures and Tables

**Figure 1 ijms-24-07037-f001:**
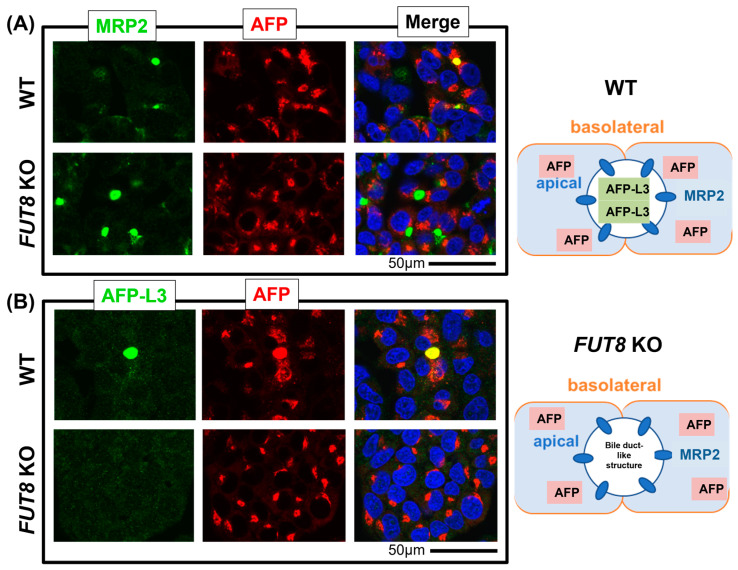
Localization of AFP-L3 was observed in the bile duct-like structures in WT HepG2 cells. (**A**) WT and *FUT8*-KO HepG2 cells were co-stained with anti-MRP2 (green) and anti-AFP (red) antibodies. (**B**) WT and *FUT8*-KO HepG2 cells were co-stained with anti-AFP-L3 (green) and anti-AFP (red) antibodies. Colocalization is displayed in yellow (Merge). Blue is the nuclear stain using 4′,6-diamidino-2-phenylindole (DAPI). The cellular localization of AFP and AFP-L3 is shown in the schematic diagram.

**Figure 2 ijms-24-07037-f002:**
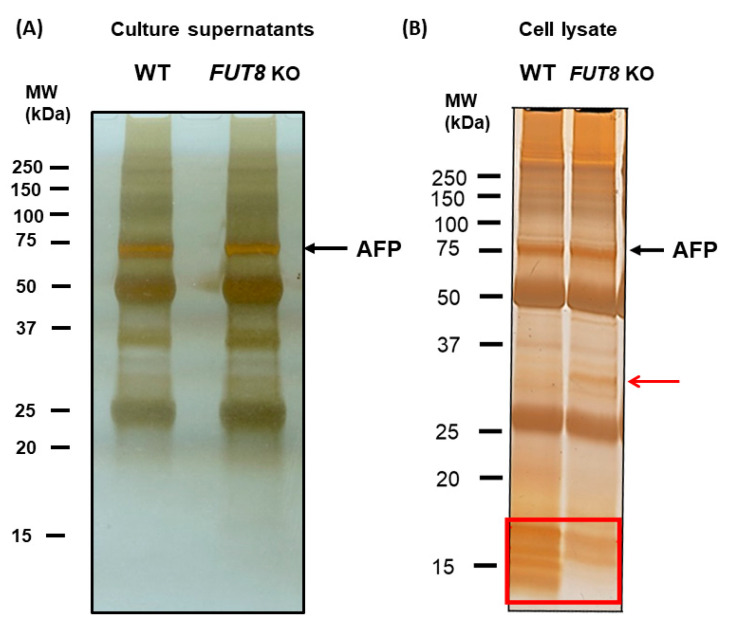
Silver staining of immunoprecipitated products from WT and *FUT8*-KO HepG2 cells. (**A**) The silver-stained gel of immunoprecipitated product with anti-AFP antibody using culture supernatants of WT and *FUT8*-KO HepG2 cells. (**B**) The silver-stained gel of immunoprecipitated product with anti-AFP antibody using cell lysates of WT and *FUT8*-KO HepG2 cells. Red arrows indicate bands specifically detected in FUT8 KO samples. Additionally, a red-lined square indicates a group of bands strongly observed in the wild type.

**Figure 3 ijms-24-07037-f003:**
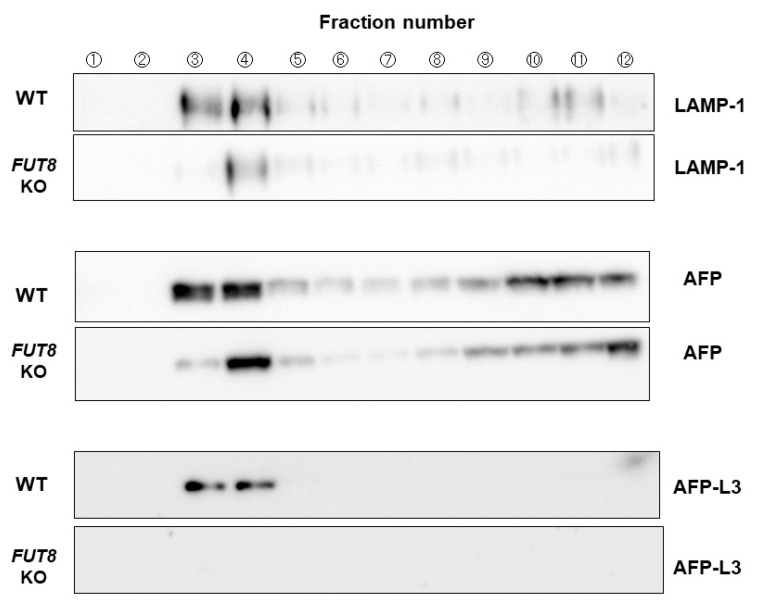
Immunoblots of endosome fractions obtained by ultracentrifugation. FUT8 KO and WT HepG2 cells were subjected to cellular fractionation by ultracentrifugation. The results of immunoblots using anti-LAMP-1, anti-AFP, and anti-AFP-L3 antibodies after electrophoresis are shown. Fractions 3 and 4 were identified as endosome fractions as they colocalized with the endosome marker LAMP-1, and they were used for subsequent immunoprecipitation experiments.

**Figure 4 ijms-24-07037-f004:**
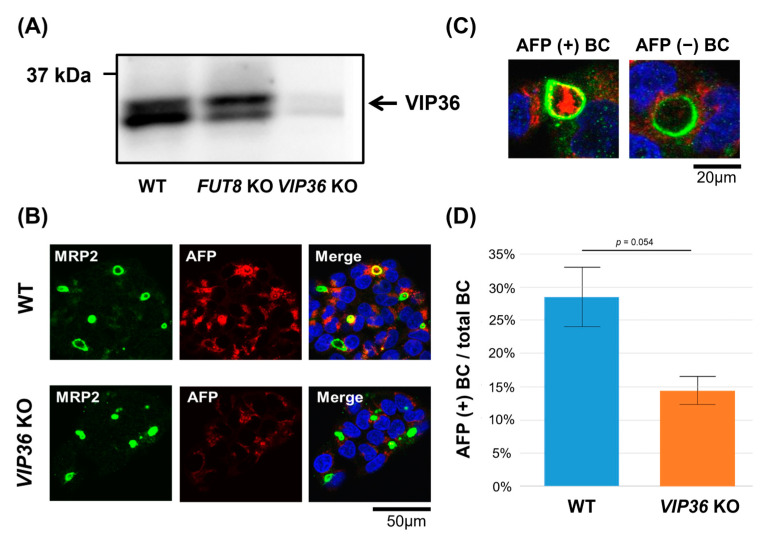
Accumulation of AFP in bile duct-like structures was reduced by *VIP36* KO in HepG2 cells. (**A**) Immunoblot of the expression level of VIP36 in *VIP36*-KO HepG2 cells compared with WT and *FUT8*-KO HepG2 cells. (**B**) Co-staining of WT and *VIP36*-KO HepG2 cells with anti-MRP2 (green) and anti-AFP (red) antibodies. Colocalization is displayed in yellow (Merge). Blue is the nuclear stain using 4′,6-diamidino-2-phenylindole. (**C**) Representative images of bile duct-like structures formed on HepG2 cells and stained with anti-AFP (red) and anti-MRP2 (green) antibodies. (**D**) The percentage of anti-AFP antibody and anti-MRP2 antibody co-positive cells in 30 randomly selected visual fields from images of WT and *VIP36*-KO HepG2 cells. BC: bile canaliculus (bile duct-like structure). Results are expressed as the mean +/− SD.

**Figure 5 ijms-24-07037-f005:**
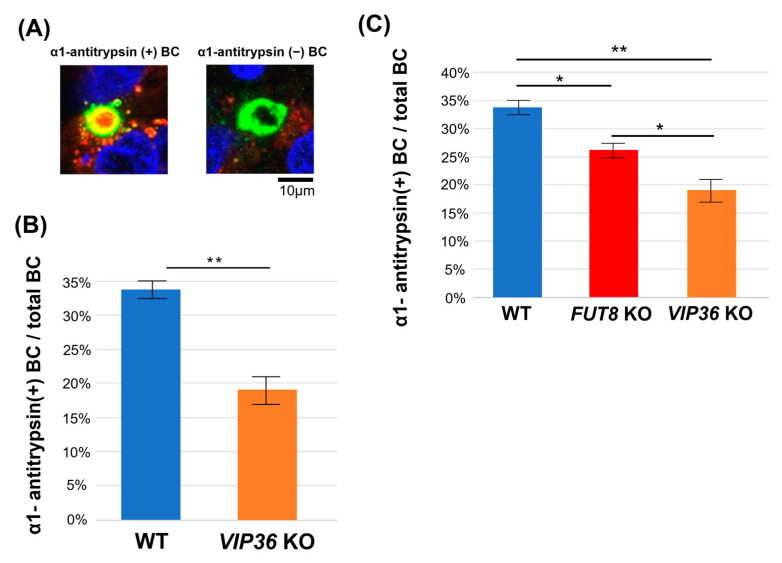
Accumulation of alpha-1 antitrypsin in bile duct-like structures was reduced by *VIP36* KO in HepG2 cells. (**A**) Representative images of bile duct-like structures formed on HepG2 cells and stained with anti-alpha-1 antitrypsin (red) and anti-MRP2 (green) antibodies. Histogram shows the percentage of anti-alpha-1 antitrypsin antibody co-positive cells in 30 randomly selected visual fields from images of WT and *VIP36*-KO HepG2 cells stained with anti-MRP2 antibody. (**B**) The percentage of anti-alpha-1 antitrypsin antibody and anti-MRP2 antibody co-positive cells in 30 randomly selected visual fields from images of WT and *VIP36*-KO HepG2 cells. (**C**) The percentage of anti-alpha-1 antitrypsin antibody and anti-MRP2 antibody co-positive cells in 30 randomly selected visual fields from images of WT, *FUT8*-KO, and *VIP36*-KO HepG2 cells. BC: bile canaliculus (bile duct-like structure). Results are expressed as the mean +/− SD. * *p* < 0.05, ** *p* < 0.01.

## Data Availability

The data that support the findings of this study are available from the corresponding author upon reasonable request.

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
