# Peer review of "Vesicular Integral-Membrane Protein 36 Is Involved in the Selective Secretion of Fucosylated Proteins into Bile Duct-like Structures in HepG2 Cells"

_ijms, 2023, doi:10.3390/ijms24087037_

Round 1

Reviewer 1 Report (New Reviewer)

     In the manuscript ijms-2281047, the authors indicate that intracellular sorting of glycoproteins are affected by their status of fucosylation. They suggested the vesicular integral-membrane protein 36 (VIP36) is the cargo-protein which is involved in the selective secretion of fucosylated proteins into bile duct-like structures in HepG2 cells.  

     The findings are very interesting but the manuscript is not well organized. I consider that this paper is acceptable for the publication in the International Journal of Molecular Sciences, after careful editing. 

Minor points

1) Abstract section is redundant and need editing. 

2) The authors should move the abbreviation HCC, from line 52 to line 50. 

3) line 66: GDP-fucose synthesis?

4) Please correct line 85. 

5) Fig. 1: Is it possible to examine the colocalization of AFP-L3 and MRP2?

6) Fig. 1: Is it possible to digitize the results just like Fig. 5 or Fig. 6?

7) Fig. 2: Is this data needed?

8) The figure legend of Fig. 3: Please indicate the origin of the sample. (WT and FUT8-KO HepG2 cells?)

9) Line 134: this sentence should be connected to the previous sentence. 

Author Response

Response to reviewer #1

 In the manuscript ijms-2281047, the authors indicate that intracellular sorting of glycoproteins are affected by their status of fucosylation. They suggested the vesicular integral-membrane protein 36 (VIP36) is the cargo-protein, which is involved in the selective secretion of fucosylated proteins into bile duct-like structures in HepG2 cells.  

     The findings are very interesting but the manuscript is not well organized. I consider that this paper is acceptable for the publication in the International Journal of Molecular Sciences, after careful editing. 

 Thank you so much for your positive comments.

Minor points

1) Abstract section is redundant and need editing. 

As a result form many time revisions, the abstract was changed from our first submission. We have amended abstract to summarize our study in the revised manuscript.

2) The authors should move the abbreviation HCC, from line 52 to line 50.           

The abbreviation of HCC is described in lines 8, page 2. We think there is no problem in this position.

3) line 66: GDP-fucose synthesis?           

We think GDP-fucose transporter is correct.

4) Please correct line 85. 

Thank you very much for pointing out our careless mistakes. We have deleted “ were identified”.

5) Fig. 1: Is it possible to examine the colocalization of AFP-L3 and MRP2?         WT

Both FasMab (anti-AFP-L3 antibody) and anti-MRP2 are mouse monoclonal antibody. It is difficult to show these co-localizations.

6) Fig. 1: Is it possible to digitize the results just like Fig. 5 or Fig. 6?     

Since there are very little co-localization of AFP and MRP-2 in Fut8 KO HepG2 cells. Therefore, we cannot count it like Fig. 5 or Fig. 6

7) Fig. 2: Is this data needed?

While VIP36 is not identified by immunoprecipitation, we think this negative data of Fig.2 is required.

8) The figure legend of Fig. 3: Please indicate the origin of the sample. (WT and FUT8-KO HepG2 cells?)         

We have added the origin of cellular samples.

9) Line 134: this sentence should be connected to the previous sentence.            

Thank you for detecting our typos. We have amended this part.

Reviewer 2 Report (New Reviewer)

This manuscript as written is very hard to follow. In many cases, it unclear why certain experiments were done and it requires guess work on the part of the reader to figure out why. In other cases, I have been unable to figure out why the experiments were done in the first place. VIP36 role as a cargo seems plausible, but lacks support by IP or chromatography. As written the manuscript in not convincing, extensive rewrite is suggested.

Introduction - Please introduce FUT8 gene, its role in fucosylation and importance in cancer. This is not obvious knowledge for non-experts. 

Figure 1 - Text in left top corner is unnecessary and is partially cut off. 

Fig 2 - Which antibody was used in IP? Figure legend says anti-AFP, but line 124 says FasMab (fucosylated AFP specific). Which one was used? Also please write clearly what is the conclusion of the experiment. It is not clear what the take home message is.

Fig 3 - Again it is very hard to follow what was done and why. Why was LAMP-1 blot done (understandably to identify endosome fractions, but this has to be explained to the readers)? AFP-L3 blot is different for WT vs FUT8-KO, why have the authors not discussed this. Again, it is hard to follow what the actual conclusion is. 

Fig 4 - What is the purpose of Fig 4A? Venn diagram in Fig 4B seems like a space filler that adds no additional value to my understanding. 

Fig 6 - It is not explained what is he relevance of alpha-1 Antitrypsin to the current study (i.e terminal fucose vs core fucose)? 

Extensive proofreading is required. There are many typographical mistakes and errors in English sentence structure.

Author Response

This manuscript as written is very hard to follow. In many cases, it unclear why certain experiments were done and it requires guess work on the part of the reader to figure out why. In other cases, I have been unable to figure out why the experiments were done in the first place. VIP36 role as a cargo seems plausible, but lacks support by IP or chromatography. As written the manuscript in not convincing, extensive rewrite is suggested.

(Answer) Thank you very much for your critical points. As you suggested, we have added several phrases to make readers understandable. If immunoprecipitation/chromatography methods can detect the interaction VIP36 and core fucose, we might find VIP36 as a cargo-protein for fucosylated glycoproteins 10 years ago. While we could not show the direct evidence of the interaction VIP36 and core fucose, we believe our findings in this manuscript will contribute to future studies for biological function of core fucose. Our results suggest that it is difficult to detect the interaction between VIP36 and core fucose by classical immunoprecipitation or chromatography methods.

.

Introduction - Please introduce FUT8 gene, its role in fucosylation and importance in cancer. This is not obvious knowledge for non-experts. 

(Answer) Thank you for critical comments. The importance of FUT8 in cancer is widely known for glycobiology researchers and summarized in many review papers such as Ref. 3 and 4. We have added this information in introduction session according to reviewer’s suggestion.

Figure 1 - Text in left top corner is unnecessary and is partially cut off. 

(Answer) Thank you for your comments. We have corrected Fig.1 in the revised manuscript.

Fig 2 - Which antibody was used in IP? Figure legend says anti-AFP, but line 124 says FasMab (fucosylated AFP specific). Which one was used? Also please write clearly what is the conclusion of the experiment. It is not clear what the take home message is.

(Answer) Thank you for your suggestion. We used anti-AFP antibody, not FasMAb. This information was described in Fig.2 legend. Although this is negative data in our experiments (another reviewer also pointed out), we would like to put in our manuscript. As you suggested, we described the conclusion of Fig.2 experiment in the revised manuscript.

Fig 3 - Again it is very hard to follow what was done and why. Why was LAMP-1 blot done (understandably to identify endosome fractions, but this has to be explained to the readers)? AFP-L3 blot is different for WT vs FUT8-KO, why have the authors not discussed this. Again, it is hard to follow what the actual conclusion is. 

(Answer) Thank you for your fruitful comments. We used LAMP-1 as a representative endosome marker and described in the Fig.3 legend. We described this information in the text of the revised manuscript. We also described the reason to perform Fig.3 experiments and we could not identify a cargo-receptor for fucosylated proteins (AFP-L3) in tmmunoprecipitation experiments using fraction 4. We have added these information in the revised manuscript.

Fig 4 - What is the purpose of Fig 4A? Venn diagram in Fig 4B seems like a space filler that adds no additional value to my understanding. 

(Answer) We are sorry that we could not understand your question. Fig.4A is raw data of silver staining of AFP-tag precipitants. Fig.4B is simple summary of proteomics experiments. If Fig.4A is not essential, we can delete this figure.

Fig 6 - It is not explained what is he relevance of alpha-1 Antitrypsin to the current study (i.e terminal fucose vs core fucose)? 

(Answer) Our previous study showed selective sorting of alpha-1 Antitrypsin with core fucose into bile is observed, using Fut8 KO mice (J Biol Chem 2006, 281, (40), 29797-806.). Therefore we used alpha-1 Antitrypsin in this study. We have added this information in the revised manuscript.

Extensive proofreading is required. There are many typographical mistakes and errors in English sentence structure.

(Answer) We are sorry that our English is not perfect. Twice native English speakers edited this manuscript and we used much cost for the editing.

Round 2

Reviewer 2 Report (New Reviewer)

The authors have sufficiently incorporated my suggestions. I suggest the manuscript for acceptance after minor changes. 

Please move Figure 4 to supplementary. It does not carry enough value to be included in the main text. 

I still found that the English required some editing. I empathize with the difficulty faced by the authors as non-native speakers and that they have undertaken a lot of expense for English proofreading. I believe research funding is better spent on science rather than on proofreading. So, I have to the best of my ability fixed the mistakes in grammar in the attached file. I hope it helps the authors. 

Author Response

Thank you very much for your positive comments and correcting English. According to your suggestion, we have changed Fig. 4 in the previous manuscript to supplemental Fig.1 in the revised manuscript. And the numbers of Fig. 5 and Fig.6 in the previous manuscript were changed as Fig. 4 and Fig. 5 in the revised manuscript, respectively.

After we make responses to another reviewer, we will have the third English proofreading by native English speakers in the revised manuscript. 

This manuscript is a resubmission of an earlier submission. The following is a list of the peer review reports and author responses from that submission.

Round 1

Reviewer 1 Report

Muranaka and coauthors report the Identification of cargo proteins including VIP36 involved in the selective secretion of fucosylated proteins into bile duct-like structures in HepG2 cell using affinity purification, mass spec proteomics and  immunostaining. The overall studies are technically sound and scientifically interesting in the field of glycobiology, cancer biomarker and immunology field and conclusions are supported by experimental evidence. 

One minor comment , in Figure 1B, it will be helpful to discuss why only a small percent of the cells appear to be AFP-L3 positive.

Author Response

Thank you very much for your positive comments

(Question) in Figure 1B, it will be helpful to discuss why only a small percent of the cells appear to be AFP-L3 positive.

Since AFP is a secretory protein, AFP with mature glycosylation such as AFP-L3 is rapidly secreted into the conditioned medium. It is possible AFP-L3 in the endosome is stained with anti-AFP-L3 antibody. However the level was very low in HepG2 cells, compared with AFP-L3 in the bile duct-like structure. However, if Huh7 cells were stained with anti-AFP-L3 antibody, positive staining of AFP-L3 was observed in the cytoplasm because Huh7 cells did not have the bile duct-like structure.

Reviewer 2 Report

In the abstract, the authors state that their proteomic study identified several candidate cargo proteins involved in the selective secretion of AFP-L3, including VIP36, a lectin-like protein that interacts with alpha-1-6-fucosylation (central fucose) on the N-glycan. Knockout of VIP36 in HepG2 cells suppressed the secretion of AFP-L3 and other fucosylated proteins, such as fucosylated alpha-1-antitrypsin, into bile duct-like structures.  And they conclude: We propose that VIP36 may be a cargo protein involved in the apical secretion of fucosylated proteins in HepG2 cells. However, in conclusion to the publication, the authors indicate that VIP36 could not be identified as a cargo receptor for fucosylated proteins, possibly due to the low affinity of VIP36 for fucosylated glycoproteins.  This dissonance between the abstract and the conclusion  left me somewhat doubtful.

More importantly, the proteomic study mentioned in the abstract, from which the authors derive their only candidate (VIP36), is not described in this article. The Strep-tag/Strep-Tactin system used here would have identified by MS about 200 proteins likely to interact with AFP in HepG2 WT 362 and FUT8-KO cells. However, the whole proteomic part, which is so important, is missing in the article, neither in the material and methods part, nor in the result part, nor in the supplemental data.

For these reasons I  strongly suggest to the authors to re-submit it with compulsory integration of their omics results with a complete description of their methods and their statistical study. As a reminder, according to the guidelines, the data provided in quantitative proteomics are obtained on the basis of biological or analytical triplicates at least. The publications below can help the authors. 

Guidelines for reporting quantitative mass spectrometry based experiments in proteomics, Bartolomé et al Journal of Proteomics 95, 16 2013, 84-88

Benchmarking Quantitative Performance in Label-Free Proteomics, James A. Dowell et al ACS Omega 2021, 6, 4, 2494–2504

Diagnostics and correction of batch effects in large-scale proteomic studies: a tutorial, Jelena ÄŒuklina et al. Mol Syst Biol, 2021;17(8):e10240

Author Response

(Comments) In the abstract, the authors state that their proteomic study identified several candidate cargo proteins involved in the selective secretion of AFP-L3, including VIP36, a lectin-like protein that interacts with alpha-1-6-fucosylation (central fucose) on the N-glycan. Knockout of VIP36 in HepG2 cells suppressed the secretion of AFP-L3 and other fucosylated proteins, such as fucosylated alpha-1-antitrypsin, into bile duct-like structures.  And they conclude: We propose that VIP36 may be a cargo protein involved in the apical secretion of fucosylated proteins in HepG2 cells. However, in conclusion to the publication, the authors indicate that VIP36 could not be identified as a cargo receptor for fucosylated proteins, possibly due to the low affinity of VIP36 for fucosylated glycoproteins.  This dissonance between the abstract and the conclusion left me somewhat doubtful.

(Answer) Thank you for your comments. We are sorry that our description was insufficient. We have changed the description, as “VIP36 could not be identified as a cargo receptor for fucosylated proteins with conventional affinity chromatography and/or immune-precipitation, perhaps owing to the low affinity of VIP36 for fucosylated glycoproteins”.

Page 25, lines 13-15 in the revised manuscript.

(Comments) More importantly, the proteomic study mentioned in the abstract, from which the authors derive their only candidate (VIP36), is not described in this article. The Strep-tag/Strep-Tactin system used here would have identified by MS about 200 proteins likely to interact with AFP in HepG2 WT 362 and FUT8-KO cells. However, the whole proteomic part, which is so important, is missing in the article, neither in the material and methods part, nor in the result part, nor in the supplemental data.

For these reasons I strongly suggest to the authors to re-submit it with compulsory integration of their omics results with a complete description of their methods and their statistical study. As a reminder, according to the guidelines, the data provided in quantitative proteomics are obtained on the basis of biological or analytical triplicates at least. The publications below can help the authors. 

(Answer) According to reviewer’s suggestion, we have added the whole results of mass-spectrometry analysis as supplemental MS data.

Page 17, lines 13-15 in the revised manuscript.

Reviewer 3 Report

This is a rather interesting article devoted to the search and study of carrier proteins for fucosylated proteins in hepatocytes. Since fucosylated proteins (including AFP-L3) are widely used as biomarkers of cancer and inflammation, the regulation of both fucosylation itself and the motility and secretion of fucosylated proteins is of great interest.

In this regard, it is very strange that there are only 18
references in the article. Moreover, more than 50% of them are articles by Japanese authors…

And it seems to me that 9 articles (or more) are self-citations!

This fact is very surprising!

The other problem I see here is that the mass spectrometry experiment is an important part of this work, and I'm sure that instead of Fig. 2C authors should provide a complete list of proteins specific to each of the cell lines in the form of a supplemental file, while a partial list of the most interesting findings should be present in the text. Because right now I don't understand why the authors choose VIP36 as the target for their experiment.

Another thing I don't understand is why the authors included the question of VIP36 pH dependence in the discussion. Since there are no experiments or quotes on this topic in the article.

Fig. 5 (graphic abstract) is not available!

Minor.

Fig. 4, B and
C 4B – the legend does not correspond to the number of graphs.
4C - the legend does not correspond to Y acis lebel.

Author Response

(Comments) This is a rather interesting article devoted to the search and study of carrier proteins for fucosylated proteins in hepatocytes. Since fucosylated proteins (including AFP-L3) are widely used as biomarkers of cancer and inflammation, the regulation of both fucosylation itself and the motility and secretion of fucosylated proteins is of great interest.

In this regard, it is very strange that there are only 18 references in the article. Moreover, more than 50% of them are articles by Japanese authors…

(Answer) Thank you for your positive comments. AFP-L3 was found by Japanese reserchers and approved by FDA in 2009. Therefore most origical studies are Japanese papers. AFP-L3 is now highly evaluated as HCC biomarker. We have cited additional 2 papers in the introdction session to show its histrory (ref. 6, 7).

Page 4, lines 9-11 in the revised manuscript

(Comments) And it seems to me that 9 articles (or more) are self-citations!
This fact is very surprising!

(Answer) It is because this manuscript is dependent on our original studies with long history after we suceeded in the purification and cDNA cloning of Fut8.

(Comments) The other problem I see here is that the mass spectrometry experiment is an important part of this work, and I'm sure that instead of Fig. 2C authors should provide a complete list of proteins specific to each of the cell lines in the form of a supplemental file, while a partial list of the most interesting findings should be present in the text. Because right now I don't understand why the authors choose VIP36 as the target for their experiment.

(Answer) Thank you very much for your critical comments. Another reviewer also make the same suggetions. According to reviewer’s suggestion, we have added the whole results of mass-spectrometry analysis as supplemental MS data. The reason for choosing VIP36 among so many proteins as cargo-receptors for fucosylated proteins is thta we found a paper “ Fiedler K. VIP36 preferentially binds to core-fucosylated N-glycans: a molecular docking study. bioRxiv. 2016: 092460”. This paper performed computed analysis of VIP36 recognition for core fucose.

(Comments) Another thing I don't understand is why the authors included the question of VIP36 pH dependence in the discussion. Since there are no experiments or quotes on this topic in the article.

(Answer) This is just our hypothesis without experomental results. Many experiments such as immunoprecopitaion and column chromatography failed to identify cargo-receptors. Therefore, we thought VIP36 pH dependence in the discussion

Fig. 5 (graphic abstract) is not available!

If so, we deleted graphic abstract.

Minor.

Fig. 4, B and
C 4B – the legend does not correspond to the number of graphs.
4C - the legend does not correspond to Y acis lebel.

(Answer) We are sorry. This is our careless mistake. We have corrected the description of Fig. 4 legend in the revised manuscript.

Round 2

Reviewer 2 Report

IN MY PREVIOUS COMMENTS, I had suggested the integration of their omics results and a complete description of their methods by reminding that according to the usual guidelines the data provided in quantitative proteomics (differential) are obtained on the basis of biological or analytical triplicates at least (this includes the statistical methods used). Here the list of proteins identified in proteomics has been provided, however with too few details concerning the number of peptides identified per protein, the number of unique and shared peptides score associated, etc... Generally the differential and discovery studies are produced on the basis of two identified peptides called proteotypic peptide to remove any ambiguity. The use of two peptides or more is interesting for increasing the confidence in the results Again, no details are provided in the publication to attest the protein identification and quantification. A complete description of the OMIC protocols and workflow and analytical methods is a prerequisite for publication. It is recognized that sample preparation individually contributes to variability, This variability in the sample preparation could lead to inconsistent results, which would make it difficult to draw meaningful conclusions from the data. It is not clear what control measures were used in this study. Have you further explored its effects by e.g. biological triplicates, as well as exploring and optimizing techniques for obtaining samples loaded on SDS-PAGE gel by technical triplicate? If so, can you integrate the results in the additional results section. In this form the article seems to be built from a single differential analysis.

Proteomics is a powerful tool in the study of proteins and their interactions in biological systems. However, the success of proteomic quantification experiments relies heavily on an accurate evaluation of the starting protein material. This is critical because the accuracy of quantification data is dependent on the quality of the sample preparation and the amount of starting protein material. Inaccurate evaluations can lead to inconsistent results, making it difficult to draw meaningful conclusions from the data. In this study, it appears that no evaluation of the protein concentration was carried out, which could have a negative impact on the results. There are many techniques available for the evaluation of protein concentrations, such as the Bicinchoninic Acid (BCA) assay or the advanced AAA assay,  that could have been used in this study. The lack of evaluation of protein concentration raises concerns about the accuracy of the results and makes it difficult to determine the reliability of the data. The SDS-PAGE gel image included in the paper also raises concerns about the quality of the sample preparation. The amounts of material loaded seem to be very different, which could indicate that the samples were not prepared consistently

In conclusion, the lack of evaluation of the protein concentration and the variability in the sample preparation raises concerns about the accuracy of the results of this study. To ensure that the results are reliable, it is important to carry out an accurate evaluation of the protein concentration and to use appropriate control measures in all proteomic quantification experiments. If the authors have this data in their possession, it would be valuable to include it in the results or in an additional data section to provide additional context for the reader.

Finally, the question that has been bothering me is this: Which cargo proteins were identified and experimentally verified in this article?

The authors state in their title and abstract, and I cite them, that several candidate cargo proteins involved in the selective secretion of AFP-L3, notably VIP36, have been identified. However, they have reached the conclusion that VIP36 cannot be considered as a cargo receptor due to its low affinity towards fucosylated glycoproteins. This leads to the question of what the other identified cargo proteins are and what role they play in the selective secretion of AFP-L3? The identification of these cargo proteins can have important implications for the understanding of the underlying mechanisms involved in the secretion process. Do the authors have any experimental data on them as they conclude that VIP36 could not be identified as a cargo receptor for fucosylated proteins. So what are these cargo proteins identified?

Author Response

As this reviewer noted, a proteomic analysis should ideally be more rigorous. In this study, mass spectrometry analysis was performed as a preliminary exploration in a setting that was not sufficiently rigorous. Also, as pointed out by another reviewer, the result in the proteomics experiment without biological triplicate is too weak to be presented as data, either for the proteins that were hit or those that were not detected. Since the main scope of this study is not the part where candidate proteins were extracted with proteomic analysis, the results and discussion independent of proteomic analysis lead to the same conclusion. Therefore, we removed the mass spectrometry analysis part from this revision.

We believe that VIP36 is the cargo protein, as indicated by the knockout experiments, but the interaction of VIP36 and core fucose might be too weak to catch with conventional methods. In the revised manuscript, we rephrased the statement from “VIP36 could not be identified as a cargo receptor for fucosylated proteins with conventional affinity chromatography and/or immunoprecipitation” to “We were unable to conclusively determine VIP36 as a cargo receptor for fucosylated proteins with conventional affinity chromatography and immunoprecipitation”.

Reviewer 3 Report

When I open pubmed, I see hundreds of articles about AFP-L3. And not all of them are from Japanese authors. Therefore, my remark about self-citation remains! Perhaps Japanese authors were the first and the best. Perhaps there are different (competing) directions. But if this is the case, the authors should explain this issue in detail.

Also, it is good that the authors have presented mass spectrometry data. But the above sup. table requires clarification.

The fact that the main result of the authors is in fact in not related to mass spectrometric analysis should also be discussed. And in this regard, the reader would be helped by a description and flowchart of the course of the entire study, which reflects what was established when the first knockout line was used, and what when the second....

My conclusion that the article should not be published in its present form.

Author Response

Thank you for your fruitful comments. As pointed out by another reviewer, the result in the proteomics experiment without biological triplicate is too weak to be presented as data, either for the proteins that were hit or those that were not detected. Since the main scope of this study is not the part where candidate proteins were extracted with proteomic analysis, the results and discussion independent of proteomic analysis lead to the same conclusion. Therefore, we removed the mass spectrometry analysis part from this revision.

The usefulness of AFP-L3 as an HCC marker was reported more than 30 years ago, and recent clinical studies and meta-analyses have accumulated evidence of its prognostic value and value in early detection. In the revised manuscript, we have cited all these clinical studies and meta-analyses published in the recent ten years according to the search in PubMed (Yi 2013 Clin Chim Acta, Cheng 2014 PloS One, Zhou 2021 Medicine, Liu 2022 Aging). Despite these clinical outputs, the underlying molecular mechanism for the increased levels of AFP-L3 in HCC patients has been unclear. We have been working on this problem for more than 15 years, which might have impressed the reviewer with “too much self-citation.” After all, we report that VIP36 is involved in the apical sorting of AFP-L3 in this manuscript. We mentioned these backgrounds in the discussion session in the secondly revised manuscript.

Round 3

Reviewer 2 Report

The major changes made by the authors make this paper publishable in the form presented. I deeply regret, nevertheless, that the proteomic part of the study was simply discarded rather than improved before the paper was re-submitted. This decision belongs to the authors, but devalues the proposed paper.

HOWEVER, the title initially given by the authors is no longer appropriate and should be modified before publication

Author Response

According to the reviewer’s suggestion, the title has been changed to “Vesicular integral-membrane protein 36 is involved in the selective secretion of fucosylated proteins into bile duct-like structures in HepG2 cells.”

Reviewer 3 Report

In the corrected version, the article looks more logical.

Minor corrections that I would suggest:

1. Divide the methods into standard subsections.

2. Slightly correct the final phrase from the 1-st chapter of the results and the title of the 3-rd chapter of the results

To identify such cargo receptors, we tried both immunoprecipitation with antibodies against AFP using a cell-conditioned HepG2 medium and cell lysates, and a study of a potential cargo receptor of core fucose described in the literature.

VIP36 knockout affects the transport of AFP into bile duct-like structures of HepG2 261 cells

Author Response

Thank you for your fruitful comments. According to the reviewer’s suggestion, we put the method session into the standard subsections and made numbers. We also deleted methods of immunoprecipitation for proteomic analyses, which we forgot to delete in the 2nd revision. We have changed the description of the result session in the 3rd revised manuscript according to the comments.
